# Differences in damping of edgewise whirl modes operating an upwind turbine in a downwind configuration

Gesine Wanke[1], Leonardo Bergami[1], and David Robert Verelst[2]

[1]Suzlon Blade Science Center, Havneparken 1, 7100 Vejle, Denmark
[2]DTU Wind Energy, Technical University of Denmark, Frederiksborgvej 399, 4000 Roskilde, Denmark

**Correspondence:** Gesine Wanke (gesine.wanke@suzlon.com)

**Abstract.** The qualitative changes in damping of the first edgewise modes when an upwind wind turbine is converted into the respective downwind configuration are investigated. A model of a Suzlon S111 2.1MW turbine is used to show that the interaction of tower torsion and the rotor modes is the main reason for the change in edgewise damping. For the forward whirl mode, a maximum decrease in edgewise damping of 39% is observed and for the backward whirl mode, a maximum increase of 18% in edgewise damping is observed when the upwind configuration is changed into the downwind configuration. The shaft length is shown to be influencing the interaction between tower torsion and rotor modes as out-of-plane displacements can be increased or decreased with increasing shaft length due to the phase difference between rotor and tower motion. Modifying the tower torsional stiffness is seen to give the opportunity in the downwind configuration to account for both, a favorable placements of the edgewise frequency relative to the second yaw frequency, as well as a favorable phasing in the mode shapes.

## 1 Introduction

Upwind wind turbines, where the rotor is placed in front of the tower relative to the wind, have been in the focus of research efforts for the recent years. As wind turbines increase in size and the cost of energy has to be reduced, rotor blades become longer and increase in flexibility. The blade tip to tower clearance is a constraint for the design of such blades. For downwind rotors, where the rotor is placed behind the tower this constraint is relaxed during normal operation and downwind rotors re-experience therefore an increase in the research effort.

The downwind concepts are known to show a higher fatigue load for the flapwise blade root moments compared to the upwind concepts due to the tower shadow effect. Glasgow et al. (1981) measured a significant fatigue load increase in the flapwise bending loads for a downwind configuration compared to an upwind configuration of a 100kW machine due to the velocity deficit of a truss tower. Zahle et al. (2009) predicted a reduction in normal force on the blade of 20%, due to the rapid fluctuation in the angle of attack as the blade passes through the tower wake. A fatigue load increase of around 20% for the damage equivalent flapwise blade root bending moment was found by Reiso and Muskulus (2013) when comparing the 5MW NREL reference turbine in a downwind configuration to the original upwind configuration.

A comparison of a full design load basis for a commercial Suzlon class IIIA 2.1MW wind turbine in an upwind configuration and a downwind configuration by Wanke et al. (2019) showed, that also the edgewise fatigue load increases significantly when

changing the upwind configuration into a downwind configuration. Only 30% of the fatigue load increase for the edgewise blade root sensor could be associated with the tower shadow effect. The remaining fatigue load increase could be associated with a lower edgewise damping in the forward whirl mode of the downwind configuration.

In the 1990s first research efforts were made to characterize the damping of the edgewise blade modes since some stall regulated turbines showed stall induced vibrations. Petersen et al. (1998a) described how the local edgewise vibrations coupled to

30 the substructure in global forward (FW) and backward (BW) whirling modes. The whirling modes resulted in a force at the hub center, rotating either with the rotational direction of the shaft (FW) or against the rotational direction of the shaft (BW). Energy was seen to be exchanged between the blade and rotor modes if the frequencies of the first edgewise FW mode and the second flapwise BW mode were placed close together. Lower damping of the modes was shown to lead to a significant increase in both fatigue and extreme loads as vibration amplitudes are higher.

In the 'STALLVIB'-project Petersen et al. (1998b) aimed to predict margins of damping, identify important parameters influencing the edgewise damping and to establish design guidelines to prevent the occurrence of stall induced vibrations. It was seen that the aerodynamic damping determined if stall induced vibrations would occur. The out-of-plane motion could generally be associated with higher aerodynamic damping. Airfoil characteristics such as the stall behavior and the slope of the lift curve over the angle of attack were found to determine if the aerodynamic force created from the vibration velocity restored

the steady-state position.

Thomsen et al. (2000) used a rotating mass on the nacelle to excite the edgewise whirling modes of a 600kW upwind turbine. From the measured blade root moment, the damping for the edgewise whirling modes was calculated. The results showed that the edgewise forward whirling mode was significantly higher damped than the corresponding backward whirling mode.

Hansen (2003) build a linearized model with 15 degrees of freedom to determine the damping for the edgewise modes of the

45 turbine, measured by Thomsen et al. (2000), using an eigenvalue approach. Hansen could confirm that the edgewise forward whirl mode was significantly higher damped than the edgewise backward whirl mode. From the visualization of the modal amplitudes, it could be shown that the edgewise forward whirl mode had a significantly higher out-of-plane component than the backward whirl mode, contributing positively to the damping. The work recommended that the overall edgewise damping could be significantly increased if the turbine design was able to place the edgewise blade frequency between the 2nd yaw and

50 tilt frequency of the turbine, as this increased the out-of-plane contribution of the edgewise forward whirl mode.

In the description of aeroelastic instabilities Hansen (2007) derived the aerodynamic damping coefficient of a single airfoil in dependency on the vibration direction. From the simplified analysis, he was able to show how the aerodynamic damping relates to the inflow velocity, the airfoil coefficients and the airfoil coefficient slopes over the angle of attack for different quadrants of vibration direction.

This paper focuses on the difference in edgewise damping when the Suzlon S111 2.1MW wind turbine is changed from an upwind configuration into a downwind configuration. The damping of the edgewise whirl modes will be estimated from time-series of the two turbine configurations and different sets of flexibility in the components. Finally, shaft length, cone angle and tower torsion are varied to show how the edgewise damping could be influenced by the turbine design.

The interaction of the rotor and the tower torsion will be shown to cause differences in the maximum damping between the two

edgewise whirl modes and the two turbine configurations. The interaction of the edgewise forward whirl mode and the tower torsion increases the edgewise damping in the upwind configuration and decreases the edgewise damping in the downwind configuration. In the forward whirl mode the edgewise damping decreases by 39% when the S111 Suzlon turbine is changed from the upwind configuration into a downwind turbine. In the backward whirl mode, the damping increases 18% when the S111 Suzlon turbine is changed from an upwind configuration into a downwind configuration. Differences in out-of-plane displacements cause the main difference in damping between the two turbine configurations and the two modes. As the eigenfrequency of the edgewise forward whirl mode is closer to the second yaw frequency the forward whirl mode will show a higher difference in damping between the configurations. The difference in damping of the forward whirl mode dominates, therefore, the overall change in damping when the upwind configuration is changed into the downwind configuration, as well as the difference in extreme and fatigue loads.

## 2 Methods

In this study, two different attempts are made to investigate the difference in edgewise damping between an upwind configuration and a downwind configuration. Firstly, the edgewise damping of the full turbine is calculated from HAWC2 (Madsen et al. (2020), Larsen and Hansen (2014) (Version 12.7)) time-series for upwind and downwind configuration with the full turbine flexibility called the fully flexible (FF) configurations. Further, the edgewise damping is estimated for turbine configurations with reduced flexibility. The flexibility is reduced by increasing the stiffness of certain turbine components significantly. The resulting configuration consists of a fully flexible rotor and a tower that is only flexible in torsion. With this configuration the difference in damping between the whirl modes and the two configurations can be captured with the minimum degrees of freedom (see also Fig. 4). The absolute damping values differ slightly between the fully flexible and the model with reduced flexibility, especially since the interaction with the tower bending is neglected. However, the model with reduced flexibility allows to separate the motion of the rotor from the motion of the substructure without the influence of nacelle or shaft. This separation allows for a more detailed root-cause analysis of the difference in damping and the visualization of out-of-plane displacement which can be attributed clearly to either rotor or tower torsional motion. The configurations with reduced flexibility are called the upwind RTT (rotor and tower torsion) and the downwind RTT configuration.

Secondly, the influence of shaft length, cone angle and tower torsional stiffness on the edgewise damping of the upwind RTT and downwind RTT configuration are studied by parametric variation. Using the RTT rotor configuration captures the trends of changes in damping with a simplified structure that allows investigations of the modal displacements. The influence of the shaft length is investigated in a range of -30% and +100%, the cone angle, coning away from the tower from $0°$ to $7.5°$ and the tower torsional stiffness in a range of $\pm 80\%$. Table 1 shows a summary of all the configurations used in the study and the investigated parameter variation. The study is based on a Suzlon S111 2.1MW, class IIIA turbine with a rotor diameter of 112m and a 90m height tubular tower. The turbine is pitch regulated and operating at variable rotor speed below rated power. The operational range is from $4\mathrm{ms}^{-1}$ to $21\mathrm{ms}^{-1}$ and rated wind speed is $9.5\mathrm{ms}^{-1}$. Blade prebend and shaft tilt are neglected in the study to reduce coupling terms between in-plane and out-of-plane modes. This assures that exclusively one of the edgewise whirl modes

**Table 1.** Configurations and parameter variations

| configuration/ parameter variation | properties |
| --- | --- |
| edgewise damping estimation | |
| all configurations | no tilt, no cone, no prebend |
| | simplified controller, no gravity, uniform inflow |
| | (no turbulence, no sheer, no veer, no inclination angle) |
| upwind FF | upwind, all degrees of freedom (fully flexible) |
| downwind FF | downwind, all degrees of freedom (fully flexible) |
| upwind RTT | upwind, rotor flexibility, tower torsion flexibility |
| downwind RTT | downwind, rotor flexibility, tower torsion flexibility |
| parameter variation | |
| shaft length | up- and downwind RTT configuration |
| | shaft length variation: -30% to +100% |
| cone angle | up- and downwind RTT configuration |
| | cone angle variation: 0° to 7.5° (away from tower) |
| tower torsional stiffness | up- and downwind RTT configuration |
| | torsional stiffness factor variation $\pm$ 80% |

is excited with the chosen method of excitation. The cone angle is neglected other than for the parameter study for the same reason. The turbine is assembled as a downwind configuration by shifting the rotor behind the tower and yawing the shaft by 180°.

## 2.1 Damping estimation from time-series

The damping of the turbine edgewise whirl modes is estimated from HAWC2 time-series. Alternatively, HAWCStab2 (Hansen (2004)) could be used to solve a linearised stability model around the non-linear deflected steady state. In doing so, the eigenfrequencies, damping and modeshapes can be obtained directly by solving an eigenvalue problem of the linearised system.

However, due to unresolved issues with respect to modelling downwind turbines in HAWCStab2 (which has only been used and tested in the traditional upwind context) it was considered outside the scope of this investigation to address those challenges. The turbine configurations from Tab. 1 are subject to a uniform wind field without turbulence, wind shear or tower shadow, to reduce the noise in the time-series. The gravity is set to zero to avoid excitation with the rotational frequency on the edgewise signal. By setting gravity to zero, the variation of stiffness is neglected in this study. Further analysis of this effect could be done

with Floquet-analysis, but this is outside the scope of this work. The controller is exchanged by a simple setting of pitch angle

and rotational speed according to the wind speed at hub height to allow for a slow wind speed increase to avoid the excitation of modal frequencies other than the edgewise whirl frequencies. This resembles a fix-free drive train operational mode. This is a simplification made in this study to avoid interaction with the controller to excite the turbine. It gives an unrealistic collective edgewise mode frequency, which is neglected in this study. It has been checked that this does not influence the asymmetric whirl modes investigated. A long run-in time is used to assure that the steady-state positions of the turbine are reached and the noise from the run-in does not disturb the vibration signal.

The forward and backward edgewise whirl modes are excited with a harmonic force at the blade at around r/R=75% radius with the blade edgewise frequency in the blade coordinate system. A time shift of 1/3 of the vibration period between the excitation forces on the three blades assures that either the forward whirl mode or the backward whirl mode is excited. The forward whirl mode is excited with the blade order 3-2-1, as the blades are named in the tower passing order seen from the front. The backward whirl mode is excited with the blade excitation order 1-2-3. After the excitation has stopped, 10 seconds of the time-series signal are neglected and 50 seconds are used for estimation of damping of the edgewise modes.

The excitation method of harmonic forcing with phase shift within the blade coordinate system would be rather difficult, if not impossible in an experimental set-up. However, it gives the opportunity in a simulation framework to excite the whirl modes separately. Figure 1 shows the spectra of the in-plane displacements for the fully flexible turbine configurations in Coleman-coordinates, obtained with the described excitation method. The figure shows that the excitation method achieves a spectrum where the symmetric component is negligible, as the amplitude significantly smaller than the amplitude of the asymmetric whirl modes. Further, Fig. 1 shows that a dominating backward ((a) and (c)) or forward whirl mode ((b) and (d)) is achieved. It is crucial for the chosen method of damping estimation that exceptionally only one of the whirl modes is excited. It has been checked that this is the case for all the regarded time-series.

From the time-series, the logarithmic damping decrement of the edgewise modes is extracted. For the estimation of damping the decaying displacement signal of the 3 blades at r/R=75% radius is used. The logarithmic damping decrement $\delta$ is calculated via

$$\delta = \frac{1}{N} ln \frac{x(t)}{x(t+NT)} \qquad (1)$$

where $N$ is the number of positive successive peaks, $x(t)$ is the edgewise displacement amplitude of the first peak and $x(t+NT)$ is the amplitude of the $Nth$ peak at $N$ vibration periods $T$ after the first peak. The logarithmic damping decrement is converted to the damping ratio $\zeta$

$$\zeta = \frac{1}{\sqrt{1 + \left(\frac{2\pi}{\delta}\right)^2}} \qquad (2)$$

The damping ratio is estimated from simulations for the backward and forward whirling mode of the fully flexible (FF) configurations as well as the upwind RTT and the downwind RTT configuration over the range of operational wind speeds. Figure 2 shows four time-series for the upwind FF configuration for the two whirl modes at 5ms$^{-1}$ and 9ms$^{-1}$ wind speed, as well as the decay function with the estimated logarithmic damping decrement. It can be seen, that generally a good estimate of damping is possible, especially if the damping is low, as shown in the time-series of 5ms$^{-1}$. As the damping increases the

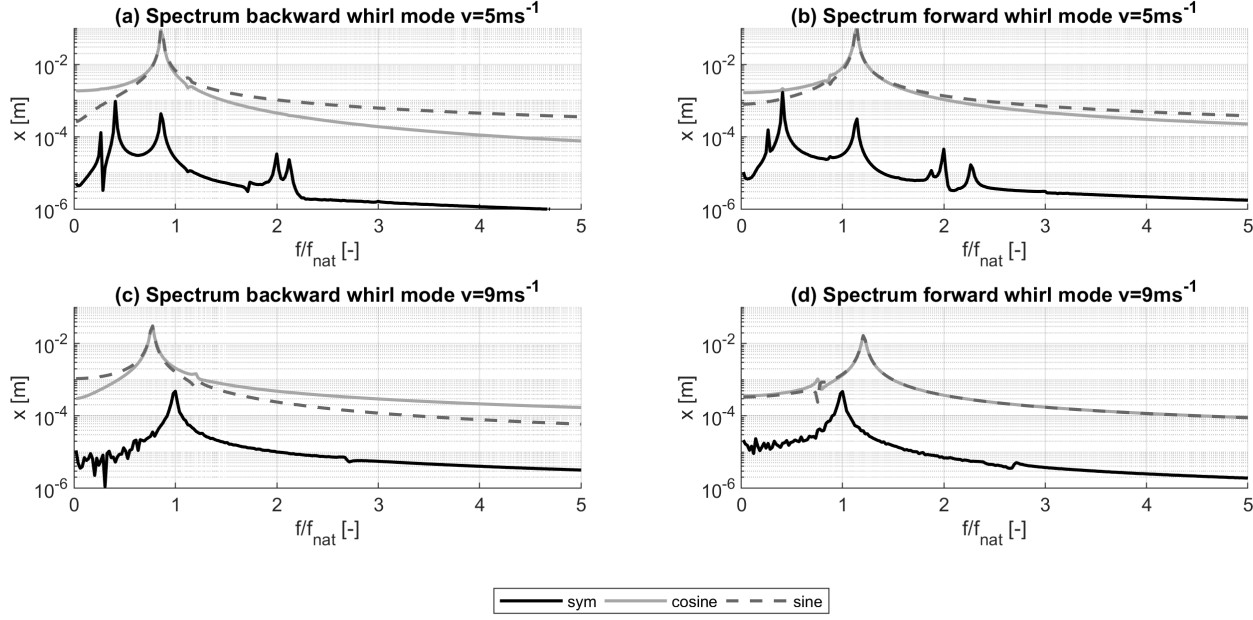

**Figure 1.** Spectrum of the in-plane displacements from time-series at $5\mathrm{ms}^{-1}$ ((a) and (b)) and $9\mathrm{ms}^{-1}$ ((c) and (d)) for the fully flexible upwind turbine configuration in Coleman-coordinates (coordinates of the fixed frame of reference, see also sect.2.2) for the backward whirl mode ((a) and (c)) and the forward whirl mode ((b) and (d)). The frequency axis is normalized with the blade edgewise natural frequency.

method becomes less accurate as the detection of the peaks becomes more difficult as seen from in the time-series of $9\mathrm{ms}^{-1}$.

The higher damping makes the peak of the spectrum broader, and more irregularities due to other frequencies than the modal frequency are cause irregularities in the time series. This approach of damping estimation from peak-to-peak counting has also been applied by Thomsen et al. (2000) under the assumption of low damped whirl modes.

As this method is using the same model as used for load simulations the method has the advantage of estimating directly the differences in damping without linearization effects. However, this method will only be able to estimate the damping, if clearly

only one mode is excited and only one frequency dominates the spectrum (see also Fig. 1). The damping has to be so low that the peak to peak counting and amplitude detection can be reliably performed. This limited effectively the investigated range of the investigated parameter. The edgewise modes are well suited for this method as they are significantly lower damped than other modes. Estimating the damping from an eigenvalue solution instead would eliminate these limitations.

The aeroelastic modal analysis tool HAWCStab2 (Hansen (2004), (Version 2.15)) has been used to calculate the eigenfrequen-

cies of the edgewise whirl mode for the excitation. HAWCStab2 has further been used to compare the trend of the damping over wind speed for both turbine configurations, as well as the ratio of damping between the upwind and the downwind configuration. The method described above is therefore assumed to be suitable for the investigation in this study.

According to Hansen (2007) the aerodynamic damping can also be estimated for a single airfoil. This primary damping esti-

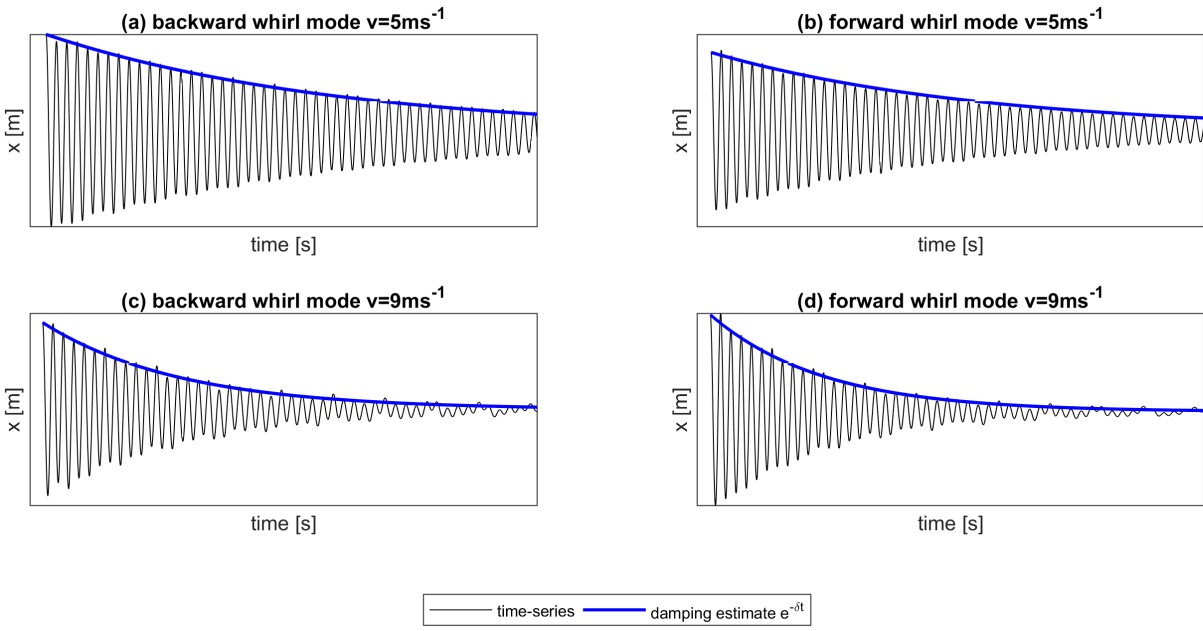

**Figure 2.** Example of estimated damping for an edgewise displacement time-series of the fully flexible upwind configuration at the backward whirl mode ((a) and (c)) and the forward whirl mode ((b) and (d)) for simulations of $5\text{ms}^{-1}$ and $9\text{ms}^{-1}$ wind speed.

mation is used to identify potential reasons for the difference in damping due to the steady-state and operational point of the airfoils of the upwind and the downwind configuration, as well as the in-plane velocity of the airfoil. The damping coefficient is calculated from simulated steady-state values of the airfoil coefficients and angle of attack at 9m/s, -3° pitch angle and r/R=75% rotor radius, as well as the respective airfoil velocity.

## 2.2 Coleman transformed time-series

By transforming the velocities and displacements to multiblade- or Coleman-coordinates (Bir (2008)) the difference in damping for the time-series at $9\text{ms}^{-1}$ can be further investigated. The time-series at $9\text{ms}^{-1}$ were chosen as the difference in damping between the two turbine configurations was observed to be largest. For the r/R=75% airfoil section of the upwind RTT and downwind RTT configuration the velocities and displacements are transformed to Coleman-coordinates such that the components due to the blade self-motion as well as the motion of the substructure can be considered. The latter is the motion of the non-deflected blade due to the tower torsion. To be able to distinguish the two components of the motion the tower torsion is monitored in the tower coordinate system, while the blade motion is monitored in the blade frame of reference. Thus, the Coleman-transformation is required to monitor all signals in the non-rotating reference frame.

A Fourier-transformation is used to calculate the spectrum of the time-series. As clearly only one mode is excited, amplitude of

the harmonic oscillation at the modal frequencies can be extracted from the spectrum for the displacement. This will be called modal displacement in this study but should not be confused with a mode shape from an eigenvector. The modal displacement and modal velocities of the r/R=75% rotor position in the Coleman-coordinates have been calculated via a discrete Fourier transformation (FFT) in Matlab (MathWorks (2020)) for the component of the rotor self-motion as well as the motion of the substructure. The FFT-analysis returns the amplitude as well as the phase of the displacements. As the phase signal is relative information for each spectrum, a reference phase signal is added to align phases from different FFT-analysis as follows:

For each displacement signal (the three Coleman-coordinate signals and the tower torsion signal) the FFT-analysis is performed, resulting in the original spectrum. In a second step, the signal of the rotational position with a factor of 1/1000 is added to the displacement signal and a second spectrum is calculated (modified spectrum). From both spectra, the phase of the edgewise whirl mode frequency is extracted. The phase signal of the modified spectrum is shifted by the difference between the two edgewise signals such that the two signals have the same phase on the edgewise whirl frequency.

$$\phi_{modified,shifted} = \phi_{modified} - \phi_{modified,mode} + \phi_{original,mode} \tag{3}$$

where $\phi_{modified,shifted}$ is the resulting shifted phase of the modified spectrum, $\phi_{modified}$ is the phase information of the modified spectrum. Both are vector information of a full spectrum, while $\phi_{modified,mode}$ and $\phi_{original,mode}$ are the phase information of the edgewise whirl mode frequency for the modified and original signal. From the shifted, modified signal the phase information at the rotational frequency is extracted. The phase information of the original signal $\phi_{original}$ is then shifted by the phase of the rotational frequency of the shifted and modified signal $\phi_{modified,shifted,rotation}$, such

$$\phi_{original,shifted} = \phi_{original} - \phi_{modified,shifted,rotation} \tag{4}$$

The shifted original signal $\phi_{original,shifted}$ aligns the phases such that the phase of the rotational signal is 0. This procedure guarantees, that the phasing between the substructure motion and the rotor motion are consistent between several FFT-calculations.

From the original, phase-shifted spectra the amplitude and phase information at the modal frequency of edgewise whirl modes is used to reconstruct the time-series of the modal displacement of the rotor as well as the modal displacement of the substructure. This is equivalent to the inverse Fourier-transformation for a single frequency of the spectrum, namely the regarded whirl frequency. The modal velocities are calculated as the derivative of the modal displacements.

## 3 Results

The result section presents the verification of the frequencies and estimated damping from HAWC2 against HAWCStab2. The section further shows the estimated edgewise damping as a function of wind speed for the fully flexible up-and downwind FF configuration as well as the up-and downwind RTT configuration. The out-of-plane displacement of the edgewise modes is shown to be the reason for the difference in damping. Finally, the damping for the parameter variation for shaft length, cone angle, and tower torsional stiffness is presented.

 ## 3.1    Comparison of whirl frequencies and damping between HAWC2 and HAWCStab2

Figure 3 shows the verification of the frequencies and the damping from the estimation in HAWC2 against the calculations with HAWCStab2 for the fully flexible turbine configurations and the two edgewise whirl modes. The figure shows that the

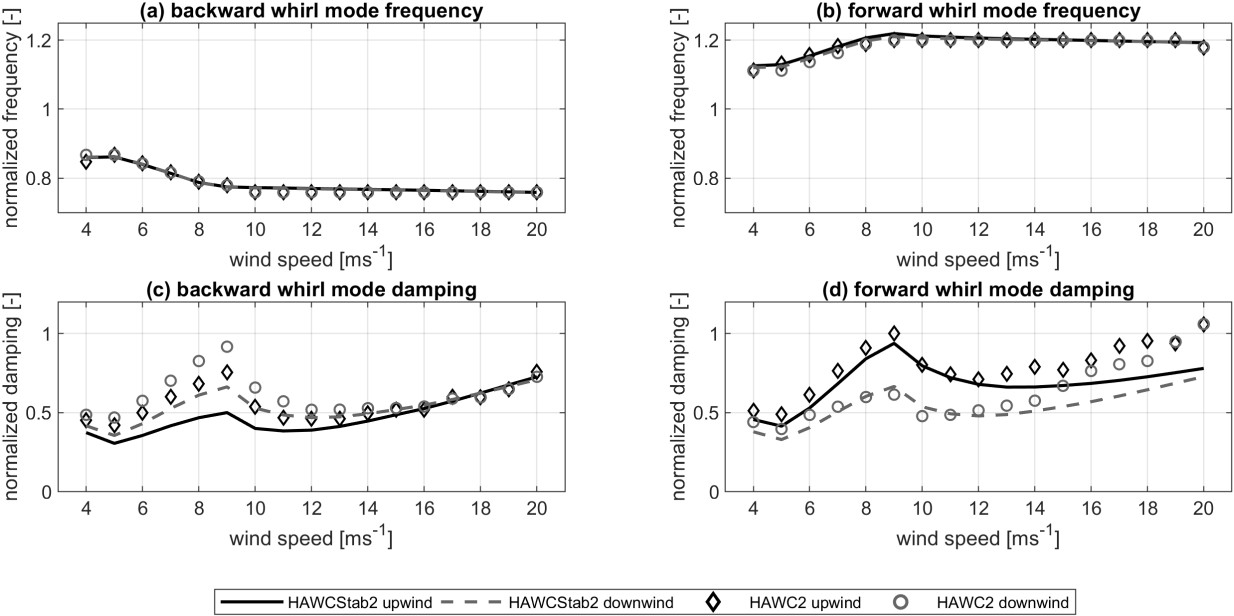

**Figure 3.** Normalized edgewise frequencies for the back whirl mode (a) and the forward whirl mode (b) for the upwind FF and the downwind FF configuration, as well as the damping of the backward whirl mode (c) and the forward whirl mode (d). The frequencies are normalized with the blade natural edgewise frequency, the damping is normalized with the damping of the upwind FF configuration at 9 ms$^{-1}$ of the forward whirl mode from the HAWC2 estimation.

frequencies calculated by HAWCStab2 and the frequencies extracted from the spectrum of HAWC2 time-series agree well. Small differences can be observed due to the resolution of the frequency axis of the spectrum. In Fig.3 (c) and (d) it can be
seen that the general change in damping over wind speed is captured by the damping estimate. Also, the changes in damping when converting an upwind configuration into a downwind configuration are captured. However, the absolute damping values estimated from the HAWC2 time-series and calculated by HAWCStab do not agree over the full wind speed range for neither configuration or whirl mode. The HAWC2 estimations predict a higher damping in the backward whirl mode around rated power than the HAWCStab2 calculations. In the forward whirl mode, the HAWC2 estimations predict higher damping for
high wind speeds than the HAWCStab2 calculations. Figure 1 ((a) and (b)) showed that the HAWC2 time-series were clearly dominated by the edgewise whirl frequencies for the fully flexible upwind configuration. Figure 2 ((c) and (d)), further showed that for the upwind configuration 9 ms$^{-1}$ the damping of both modes should be equally well estimated from HAWC2 time-

series. This comparison indicates that more investigations are required to determine the reason for the differences in damping between the two methods. However, as the differences in damping between the turbine configurations are captured, the method

of damping estimation from HAWC2 is assumed to be suitable for further investigations of damping difference in the edgewise whirl modes between the upwind and the downwind configuration.

The attempt of comparing the mode shapes has not been successful, as the reconstructed modal phases from the available HAWCStab2 output showed inconsistencies. The reason for those inconsistencies could not be found and further investigations are needed.

**3.2 Edgewise damping over wind speed estimated from time-series**

Figure 4 shows the estimated normalized damping ratio as a function of wind speed for the backward (Fig. 4 (a)) and forward whirl mode (Fig. 4 (b)) for the fully flexible up- and downwind FF configuration, as well as the upwind RTT and downwind RTT configuration. The figure further shows the difference in damping between the upwind configuration RTT and the other configurations (Fig. 4 (c) and (d)). Both, the damping as well as the damping difference are normalized with the damping of

225 the upwind RTT configuration at $9 \text{ ms}^{-1}$ of the forward whirl mode. The figure shows that both edgewise whirl modes in both configurations are positively damped. The damping ratio increases from cut-in wind speed to a local maximum at rated wind speed. After decreasing for wind speeds between rated wind speed and wind speeds around $14 \text{ms}^{-1}$, a damping increase for wind speeds higher than $14 \text{ms}^{-1}$ is observed. In the backward whirl mode (Fig. 4 (a)) the downwind configurations are subject to higher edgewise damping than the respective upwind configurations. The difference is approximately 18% for the

230 RTT configurations (see Fig. 4 (c)). In the forward whirling mode (Fig. 4 (b)) the two downwind configurations are subject to significantly lower edgewise damping than the respective upwind configurations over the investigated wind speed range. The difference in edgewise damping is largest around rated wind speed, where the damping is approximately 39% lower in the downwind RTT configuration than the upwind RTT configuration (Fig. 4 (d)).

Generally the RTT configurations show the same trend over wind speed as the FF configurations. Also the differences between

235 the upwind and the downwind configuration is captured by the RTT configurations. Only in the backward whirl mode of the downwind configuration the change in damping over wind speed is stronger pronounced for the RTT configuration than for the FF configuration. Thus, the degrees of freedom neglected with the RTT rotor configuration do impact the upwind and the downwind configuration in the same manner and are not a major causation of the difference in edgewise damping between the two turbine configurations. The RTT configuration has the advantage that modal displacements can be investigated in more

detail without the influence of shaft or nacelle motion. The RTT configuration is therefore used for the parameter study. For the upwind configurations the forward whirl mode (Fig. 4 (b)) is significantly higher damped than the backward whirl mode (Fig. 4 (a)), as also shown by Hansen (2003), since the forward whirl mode has a higher out-of-plane component of the mode shape than the backward whirl mode. When the tower flexibility is removed from the model or when the aerodynamic forces are not present, the damping of both forward and backward modes are identical (not shown in Fig. 4). This indicates that the difference

in whirl damping between the upwind and the downwind configuration is driven by the interaction of the aerodynamic forces on

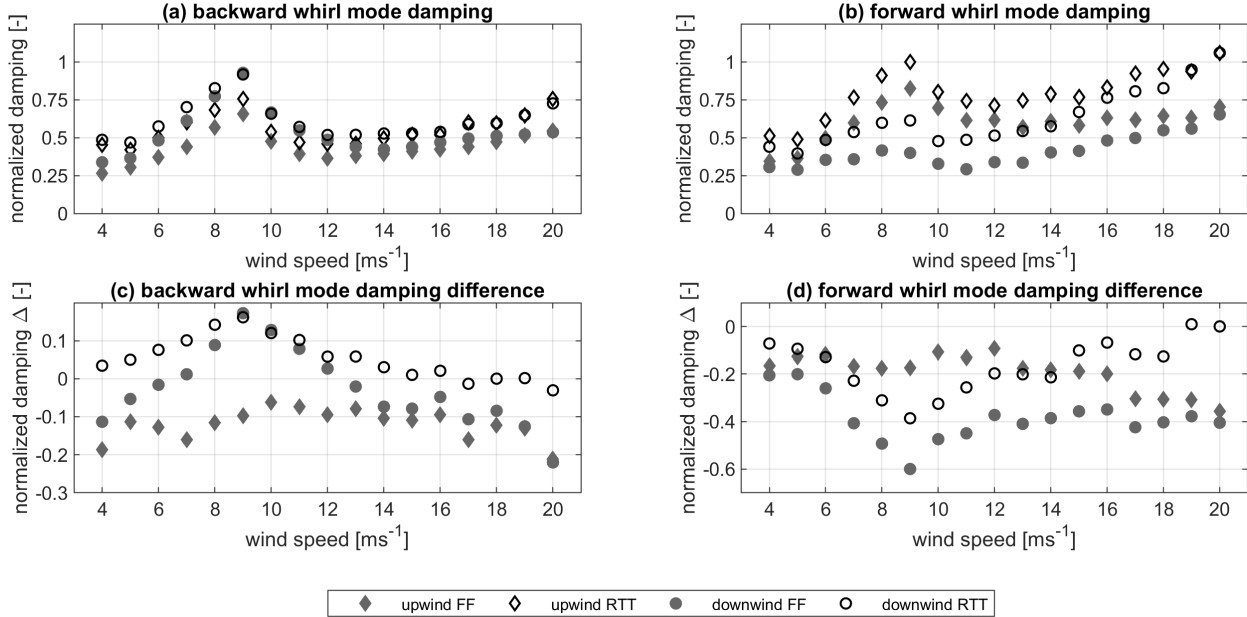

**Figure 4.** Normalized damping ratio as function of wind speed for the edgewise backward whirl mode (a) and the forward whirl mode (b) for the upwind RTT and the downwind RTT configuration and the fully flexible FF configurations, as well as the difference in damping to the damping to the upwind RTT configuration in the backward whirl mode (c) and the forward whirl mode (d). A positive difference indicates a larger damping than for the upwind RTT configuration. The damping, as well as the damping difference, are normalized with the damping of the upwind RTT configuration at 9 ms$^{-1}$ of the forward whirl mode.

the rotor with the tower torsional motion. Other degrees of freedom such as tower bending flexibility do influence the absolute damping of the whirl modes but do not cause the major differences between the upwind and the downwind configuration.

### 3.3 Modal displacement effects on edgewise damping

The observed difference in normalized edgewise damping between the upwind and the downwind configuration presented in
Fig. 4 cannot be explained with the pure in-plane damping coefficient of an airfoil section as modelled analytically by Hansen (2007). There is no difference in the coefficient since the steady-state values of the airfoil coefficients and the respective slopes are the same. Further, no difference in the in-plane velocities could be found. Thus, the difference in edgewise damping has to be explained by the out-of-plane displacements in the modes for the different turbine configurations. To simplify the analysis the RTT configuration is considered, where the out-of-plane displacements can only be either due to the flap component in the
edgewise modes or due to the tower torsion that rotates the blades out of the reference plane. This section shows

- higher out-of-plane displacement gives higher edgewise damping

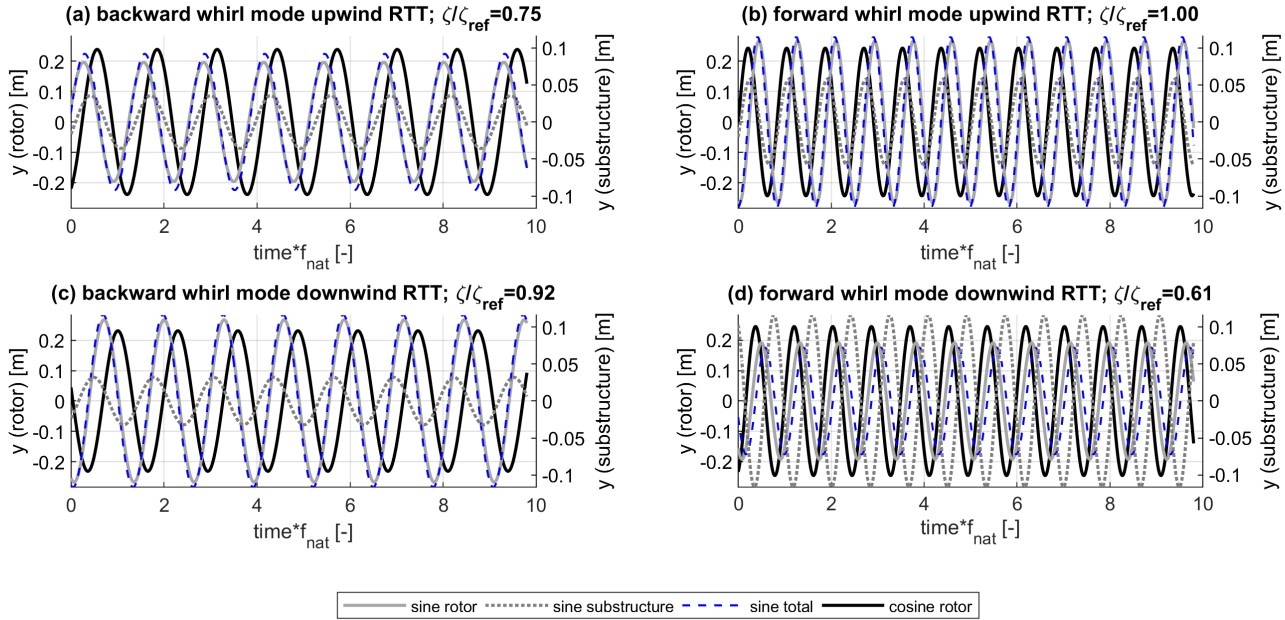

**Figure 5.** Modal out-of-plane displacements at $9\,\mathrm{ms}^{-1}$ for the backward whirl mode ((a), (c)) and forward whirl mode ((b), (d)) of the upwind RTT ((a), (b)) and downwind RTT configuration ((c), (d)). The time axis is normalized with the blade edgewise natural frequency.

- difference in forcing due to configuration gives a difference in modal phases

- difference in modal phases gives a difference in damping

Figure 5 shows the out-of-plane displacements of the sine and cosine components of the rotor, as well as the sine displacement
component due to the substructure, e.g. the tower torsion, for the backward whirl mode (Fig. 5 (a) and (c)) and the forward whirl mode (Fig. 5 (b) and (d)) of the upwind RTT and the downwind RTT configuration.

The figure shows that there generally is a phase shift between the sine component of the out-of-plane displacement between the substructure and the rotor. Adding the two signals leads to the total sine component of the out-of-plane displacement with the same frequency, but a different amplitude and phase. Only in the forward whirl mode of the downwind RTT configuration,
which is also the mode with the overall lowest damping, the total out-of-plane displacement is reduced due to the tower torsional displacement (Fig. 5 (d)). Generally, the main contribution to the out-of-plane displacement is due to the rotor self-motion. The forward whirl mode (Fig. 5 (b) and (d)) shows generally higher out-of-plane displacements of the substructure than the respective backward whirl mode (Fig. 5 (a) and (c)), as the natural frequency of the forward whirl mode is closer to the natural frequency of the second yaw mode. The natural frequencies of the modes are the same for the upwind and the
downwind configuration.

The interaction of the rotor and the tower causing a higher sine out-of-plane displacement of the rotor leading to higher

damping in the forward whirl mode of the upwind RTT configuration (5 (b)) and the backward whirl mode in the downwind RTT configuration (5 (c)) than respective modes with lower sine out-of-plane displacements. The forward whirl mode of the upwind RTT configuration (5 (b)) further shows a 5% higher out-of-plane cosine component of rotor displacement than the downwind RTT configuration in the backward whirl mode (5 (c)), which explains the remaining difference in damping between the two turbine configurations.

The higher sine component of the rotor out-of-plane displacement cannot be associated with the frequency of the second yaw mode, as this does not hold true for the backward whirl mode of the downwind configuration (Fig. 5 (c)). The higher out-of-plane rotor displacement in the sine component is observed to come with a sine component of the substructure out-of-plane displacement that is lagging the respective rotor displacement (Fig. 5 (b) and (c)). If the sine component of the out-of-plane displacement of the substructure is leading the respective rotor displacement the sine component of the out-of-plane rotor displacement is lower (Fig. 5 (a) and (d)).

Also the in-plane motion of the rotor (not shown here) is subject to a sine component lagging the cosine component in the backward whirl mode and a sine component leading the cosine component in the forward whirl mode. The modal velocities cause aerodynamic forces. Inherently to the whirl modes the aerodynamic in-plane forces at the hub sum up to a non-zero total in-plane force. With the arm of the shaft length, this force causes a yaw loading. Depending on the placement of the rotor relative to the yaw center a positive in-plane cosine force at the hub causes a positive yaw loading (upwind configuration) or a negative yaw loading (downwind configuration). The response of the tower, e.g. the out-of-plane substructure sine component of displacement is therefore either lagging the out-of-plane sine component of the rotor displacement, as in the forward whirl mode of the upwind RTT configuration (5 (b)) and the backward whirl mode of the downwind RTT configuration (5 (c)) or the sine component of the substructure out-of-plane displacement is leading the rotor sine out-of-plane displacement (upwind RTT configuration, backward whirl mode, Fig. 5 (a) and downwind RTT configuration, forward whirl mode, Fig. 5 (d)).

From this analysis, it can be seen that the difference in edgewise whirl damping is due to a difference in total out-of-plane motion. The main contributor is the higher rotor out-of-plane motion associated with a favorable phasing between the out-of-plane motion of the substructure and the out-of-plane sine component of the rotor motion. It will, therefore, be expected from the analysis described in the previous paragraphs, that the edgewise whirl damping can be increased by an increase of the yaw loading if the substructure displacement is lagging the sine out-of-plane displacement of the rotor. The edgewise whirl damping is on the other hand expected to decrease with an increased yaw loading if the substructure displacement is leading the sine out-of-plane displacement of the rotor. Increasing the shaft length is expected to increase the damping of the edgewise forward whirl mode in the upwind configuration as well as in the backward whirl mode of the downwind configuration. In the backward whirl mode of the upwind configuration and the forward whirl mode of the downwind configuration, an increase in shaft length is expected to decrease the edgewise damping.

### 3.4   Parameter variation

The following sections present the effect of shaft length, cone angle and tower torsional stiffness on the edgewise whirl damping. The study is performed on the RTT configuration as it allows a simplified investigations of the out-of-plane displacements.

However, since other important structural interactions such as tower bending are neglected the presented mapped edgewise whirl damping only show the trends of changes of damping in relation to the varied parameter. The effect of the single parameter are emphasised and can not be compared between each other.

### 3.4.1 Shaft length

Figure 6 shows the normalized edgewise whirl damping values of the backward whirl mode (Fig. 6 (a) and (c)) and forward whirl mode (Fig. 6 (b) and (d)) for the upwind RTT configuration (Fig. 6 (a) and (b)) and the downwind RTT configuration (Fig. 6 (c) and (d)) as a function of wind speed and shaft length factor. The figure shows that the normalized edgewise whirl

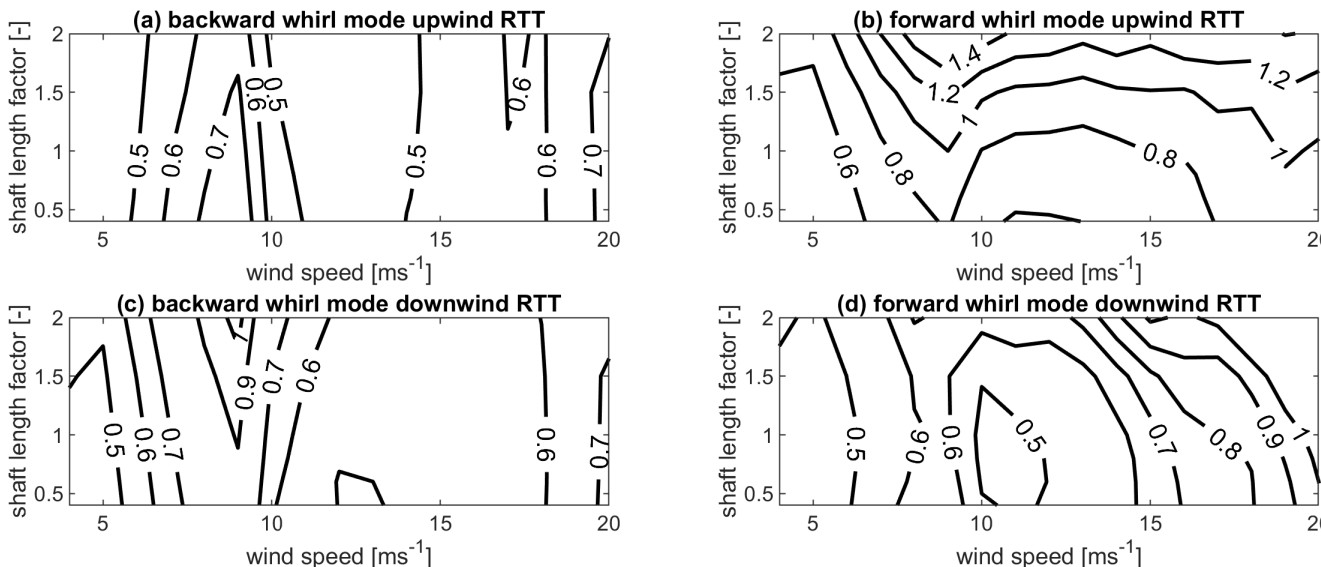

**Figure 6.** Normalized edgewise whirl damping ratio as a function of wind speeds and shaft length factor for the backward whirl mode ((a) and (c)) and the forward whirl mode ((b) and (d)), for the upwind RTT configuration ((a) and (b)) and the downwind RTT configuration ((c) and (d)). The edgewise whirl damping is normalized with the damping value of forward whirl mode in the upwind RTT configuration at a shaft length factor of 1 at $9\mathrm{ms}^{-1}$.

damping of the backward whirling mode in the upwind RTT configuration decreases with the increasing shaft length (Fig. 6 (a)). In the downwind RTT configuration, on the other hand, the normalized edgewise whirl damping increases with the increasing

shaft length in the backward whirl mode (Fig. 6 (c)). The effect is strongest pronounced around rated wind speed. In the forward whirl mode of the upwind RTT configuration, the normalized edgewise damping increases with the increasing shaft length (Fig. 6 (b)). The normalized edgewise whirl damping of the downwind RTT configuration on the other hand hardly changes with the increasing shaft length for wind speeds close to rated wind speed (Fig. 6 (d)).

For a shaft length factor of 2 and at $9\mathrm{ms}^{-1}$ the out-of-plane displacements (see appendix Fig. A1) have been compared to the

320 displacements for a shaft length factor of 1 at $9\mathrm{ms}^{-1}$ (Fig. 5). Both turbine configurations in the backward whirl mode and

also the upwind RTT configuration in the forward whirl mode show the expected dependence on the shaft length according to Sect. 3.3 around rated wind speed: the normalized edgewise whirl damping of the backward whirl mode of the downwind RTT configuration and the normalized edgewise whirl damping of the forward whirl mode in the upwind RTT configuration are increasing due to higher out-of-plane displacements in the rotor sine components. In the backward whirl mode of the upwind RTT configuration, a decrease of the sine component of the out-of-plane rotor displacements can be observed. Also for the downwind RTT configuration in the forward whirl mode, the expected decrease of out-of-plane sine component of rotor displacement can be observed. However, an increase of the cosine out-of-plane displacements of the rotor can also be seen. The combination of the out-of-plane displacements leads to the effect that hardly any difference in edgewise whirl damping can be observed at around rated wind speed for the forward whirl mode of the downwind RTT configuration.

## 3.5 Cone angle

Figure 7 shows the normalized edgewise whirl damping for the backward whirl mode (Fig. 7 (a) and (c)) and the forward whirl mode (Fig. 7 (b) and (d)) in the upwind RTT (Fig. 7 (a) and (b)) and the downwind RTT configuration (Fig. 7 (c) and (d)) as a function of cone angle and wind speed.

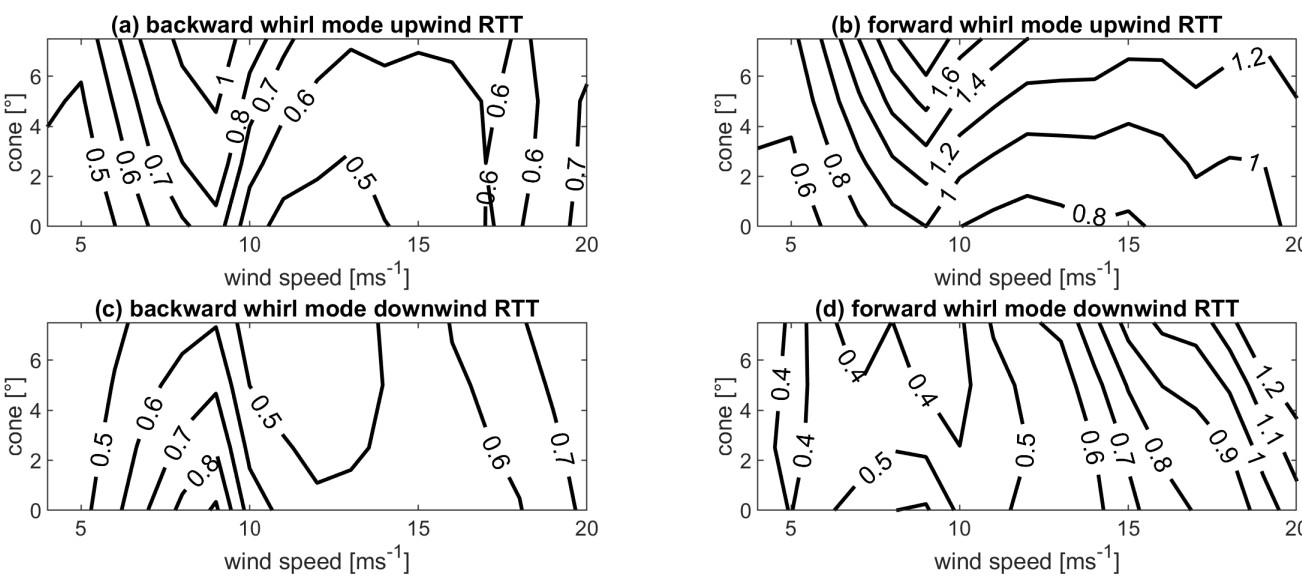

**Figure 7.** Normalized edgewise whirl damping as a function wind speeds and cone angle for the backward whirl mode ((a) and (c)) and the forward whirl mode ((b) and (d)), for the upwind RTT configuration ((a) and (b)) and the downwind RTT configuration ((c) and (d)). The whirl damping is normalized with the damping value of forward whirl mode in the upwind configuration at $0°$ at 9 ms$^{-1}$.

The figure shows that the edgewise whirl damping of both modes increases with an increasing cone angle in the upwind RTT configuration (Fig. 7 (a) and (b)). In the downwind RTT configuration the edgewise whirl damping decreases with increasing cone angle for wind speeds around rated wind speed (Fig. 7 (c) and (d)).

Introducing the cone angle has several effects. On the one hand, the cone angle changes the steady-state values of the airfoil coefficients and therefore the estimated analytical damping coefficient by Hansen (2007): The blades deflect against the coning direction in the upwind RTT configuration, while the blades deflect in the same direction as the cone direction in the downwind RTT configuration. The analytical damping coefficient of the r/R=75% airfoil at $9ms^{-1}$ has decreased by 33% in the upwind RTT configuration when a cone angle of 7.5° is introduced. The analytical damping coefficient of the r/R=75% airfoil at $9ms^{-1}$ has increased by 38% in the downwind RTT configuration when a cone angle of 7.5° is introduced. On the other hand, the cone angle also changes the coupling between the in-plane loading and the tower torsion as the distance between the yaw axis and the outboard airfoils is increased. Comparing the displacements at $9ms^{-1}$ at a cone angle of 7.5° (Fig. A2) with the out-of-plane displacements at $9ms^{-1}$ without cone angle (Fig. 5) shows only very little changes in the rotor sine components of the out-of-plane displacements. However, the downwind RTT configuration shows a significant decrease in the cosine component of the out-of-plane rotor displacements, while the upwind RTT configuration shows a significant increase in the cosine out-of-plane rotor displacements when the cone angle of 7.5° is introduced. The changes in the cosine out-of-plane rotor displacement dominate the change in normalized edgewise whirl damping.

## 3.6 Tower torsion

Figure 8 shows the normalized edgewise whirl damping for the backward whirl mode (Fig. 8 (a) and (d)) and forward whirl mode (Fig. 8 (b) and (c)) of the upwind RTT (Fig. 8 (a) and (b)) and downwind RTT configuration (Fig. 8 (c) and (d)) as a function wind speed and tower torsion stiffness factor.

While the normalized edgewise whirl damping is increasing in the backward whirl mode of the upwind RTT configuration with the increasing tower torsional stiffness (Fig. 8 (a)), the normalized edgewise damping of the backward whirl mode of the downwind RTT configuration (Fig. 8 (c)) is decreasing with increasing tower torsional stiffness at around rated wind speed. For high wind speeds and stiffness factors lower than 0.5 the edgewise damping of the backward whirl mode increases drastically with the decreasing tower torsional stiffness for both configurations. In the forward whirl mode, the normalized edgewise damping generally decreases in both configurations with an increasing tower torsional stiffness (Fig. 8 (b) and (d)). Both configurations in the forward whirl mode show an area at around cut-out wind speeds at a stiffness factor at around 0.5, where a local maximum of normalized edgewise damping is reached.

Comparing Fig. A3 with Fig.5 shows that a decrease of tower torsional stiffness to a factor of 0.2 at $9 \text{ ms}^{-1}$ increases generally the out-of-plane displacements associated with the substructure. Further, phasing between the substructure and rotor out-of-plane displacement as well as the rotor associated out-of-plane displacement is changing. Overall, only the upwind RTT configuration in the backward whirl mode does not benefit from the decrease in the tower torsional stiffness in the out-of-plane displacements at $9 \text{ ms}^{-1}$. At high wind speeds, the effect of the frequency placement can be observed. As the tower torsional stiffness decreases below a factor of around 0.5 the second yaw frequency crosses the edgewise forward whirl mode frequency and moves closer to the edgewise backward whirl mode frequency. Thus, the highest damping at high wind speeds is observed at a stiffness factor of around 0.5 for high wind speeds.

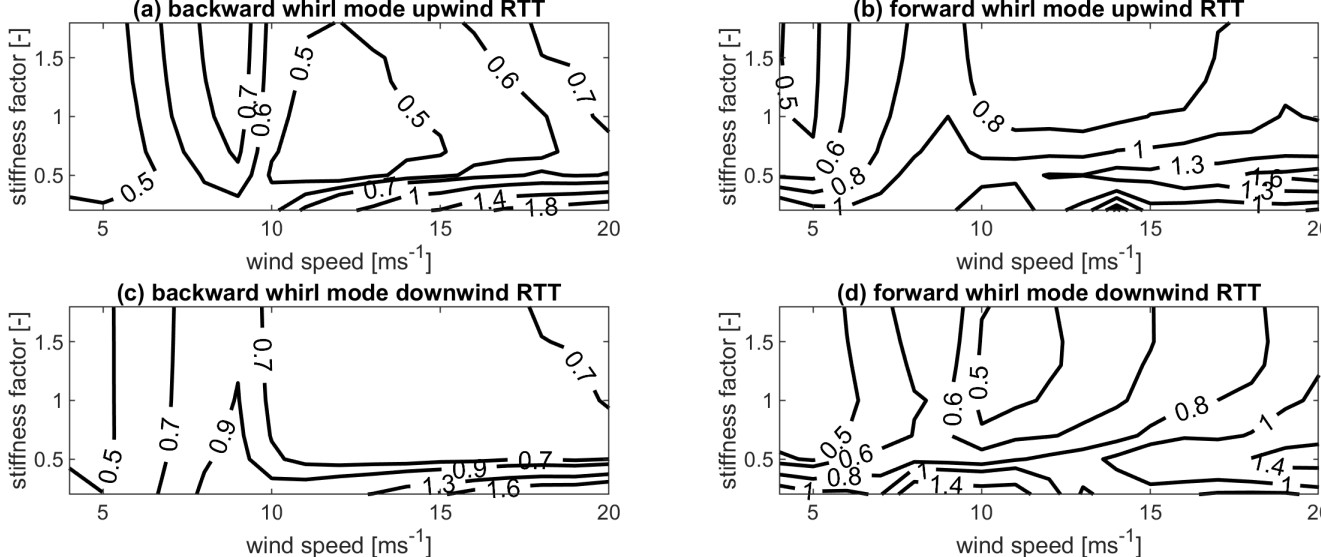

**Figure 8.** Normalized edgewise whirl damping as a function of wind speeds and tower torsion stiffness factor for the backward whirl mode ((a) and (c)) and the forward whirl mode ((b) and (d)), for the upwind RTT configuration ((a) and (b)) and the downwind RTT configuration ((c) and (d)). The whirl damping is normalized with the damping value of forward whirl mode in the upwind RTT configuration at a tower torsion stiffness factor of 1 at 9 ms$^{-1}$.

## 4  Summary

In this article, the change in edgewise whirl damping when an upwind wind turbine is converted into a downwind configuration has been investigated on the example of a simplified version of the commercial Suzlon S111 2.1MW wind turbine. The edgewise forward whirl mode has been shown to decrease in damping as the upwind configuration is changed into the downwind configuration. The edgewise backward whirl mode, on the other hand, has been seen to increase in damping when the upwind configuration is changed into a downwind configuration. The interaction with the aerodynamic forces, the rotor and tower torsional motion have been shown to create a difference in out-of-plane displacement. The out-of-plane displacement was seen to cause the observed differences in edgewise damping.

The difference in the out-of-plane displacements and therefore damping was shown to increase with an increased shaft length, as the yaw loading from the in-plane cosine shear forces could be increased. An increase in cone angle has been shown to increase the cosine component of the out-of-plane rotor displacements and therefore damping for the upwind configuration, while the increase in cone causes a decrease in cosine component of the out-of-plane displacements and damping in the downwind configuration. A decrease in tower torsional stiffness has been seen to increase the damping from a favorable placement of natural frequencies relative to each other, as long as the rotor and substructure out-of-plane displacement do not counteract each other due to phase differences.

## 5 Conclusion and future work

As an over all decrease in the whirl damping increases extreme as well as fatigue loads, the edgewise damping should be included in the design considerations. For the shaft length, there would be a trade-off between edgewise damping of the two modes, but also the rotor overhanging moment that has to be carried by the support structure. The consideration of edgewise damping would suggest a higher cone angle for upwind configurations than for downwind configurations. Again, other considerations like tower clearance, flapwise blade root loads and power production compete in the design decision. From an edgewise damping point of view downwind configurations could benefit from towers with lower torsional stiffness. Replacing a tubular tower or the bottom segments of the tubular tower by a lattice structure could significantly increase the overall edgewise damping. However, a full load assessment is required to also investigate the influence on other load sensors than the edgewise blade loads.

It should be pointed out, that the damping of the edgewise whirl modes and especially the mode shapes are turbine specific. However, it can generally be concluded that upwind and downwind configurations are expected to have different out-of-plane displacements due to the structural arrangement of the tower center relative to the rotor center. The damping of the backward whirl mode is expected to increase while the damping of the edgewise forward whirl mode is expected to decrease due to the change in out-of-plane displacement when the upwind configuration is changed into a downwind configuration. A test of the edgewise whirl damping of the DTU-10MW-RWT confirmed the same trends of difference in damping for the two edgewise whirl modes when the upwind configuration is changed into a downwind configuration. Generally, the tower torsion will have a great impact on the difference in edgewise whirl damping between the upwind and the downwind configuration. Tower torsional stiffness, cone angle and shaft length will be able to influence the difference in edgewise whirl damping. The impact of each design parameter is expected to vary from turbine to turbine, individually as well as relative to each other as it depends on the individual turbines mode shapes and frequency placement.

The damping of the first two edgewise whirl modes has been estimated from time-series where the forward or backward whirl mode is excited. The chosen example turbine was well suited for the investigations since the edgewise whirl modes are significantly lower damped than other modes and no other frequency was placed close by the edgewise whirl frequencies. Estimating the damping from an eigenvalue solution would eliminate these limitations and broader investigations also regarding other turbines classes and rated power could be made.

Further, the study of out-of-plane displacements should be extended to the turbine model with full flexibility, as additional degrees of freedom are expected to affect the mode shapes, especially the turbine tilting flexibility (tower fore-aft and shaft bending flexibility), or shaft bending are expected to influence the out-of-plane displacements. The tower side-side deflection can be expected to couple directly to the asymmetric edgewise motion and therefore influences the mode shapes. The tower side-side deflection could further be expected to couple to out-of-plane displacement through the yaw motion, as the center of gravity of the rotor is not at the tower center. Also, the shaft torsional flexibility could influence the edgewise damping. Further investigations should evaluate the influence of the full turbine flexibility especially on the mode shapes of the whirl modes.

Future work should investigate further the reason for the different out-of-plane displacement in the mode shapes, especially

the differences observed in the cosine components of the out-of-plane displacement. Gaining a full understanding of the out-of-plane displacement is important as it enables to design wind turbines with higher damping and therefore lower fatigue and extreme loads.

*Data availability.* The data is not publicly accessible, since the research is based on a commercial turbine and the data is not available for disclosure by Suzlon.

## Appendix:  Out-of-plane displacements for parameter variations

The following figures show the out-of-plane displacements of the two edgewise damping modes in the two RTT configurations at 9ms$^{-1}$ for a shaft length factor of 2 (Fig. A1), a cone angle of 7.5° (Fig. A2) and a tower torsional stiffness factor of 0.2 (Fig. A3).

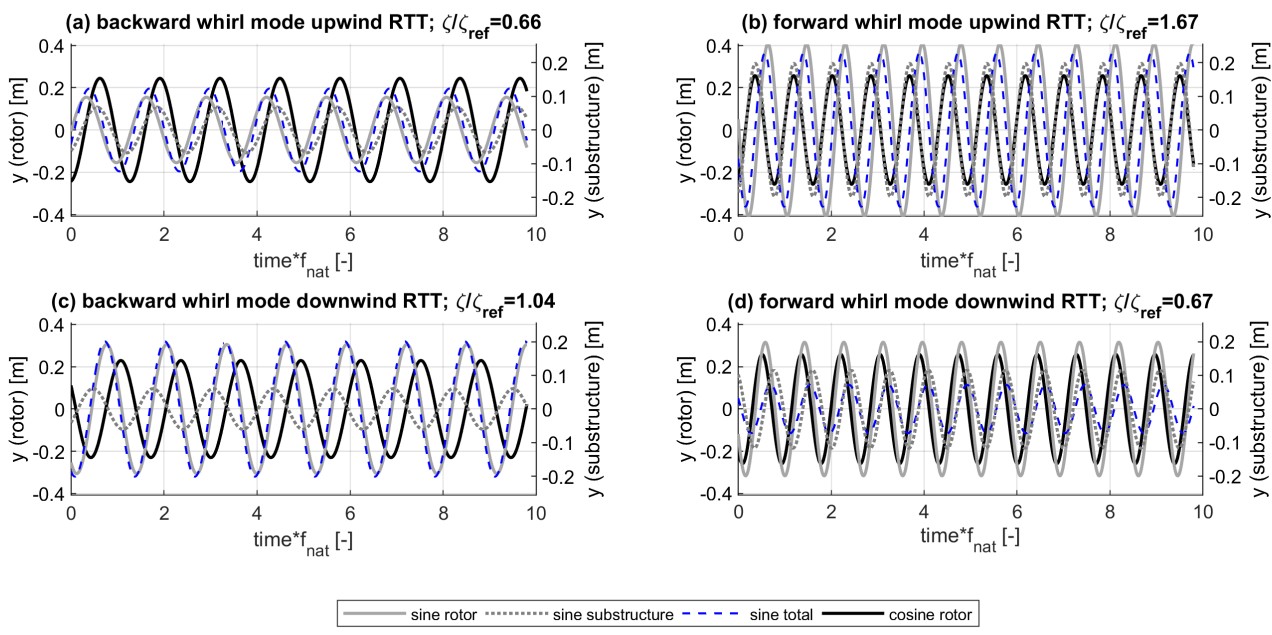

**Figure A1.** Modal out-of-plane displacements at $9\text{ms}^{-1}$ and a shaft length factor of 2 for the backward whirl mode and forward whirl mode of the upwind RTT and downwind RTT configuration. The time axis is normalized with the blade edgewise natural frequency.

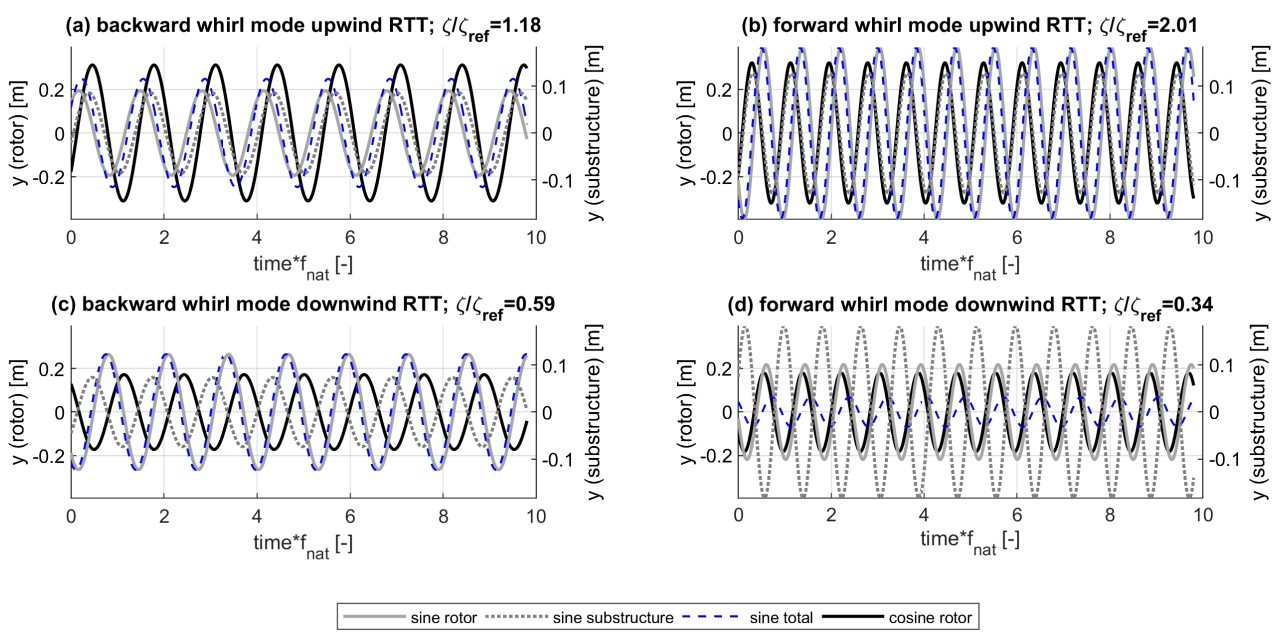

**Figure A2.** Modal out-of-plane displacements at $9\mathrm{ms}^{-1}$ and a cone angle of $7.5°$ for the backward whirl mode and forward whirl mode of the upwind RTT and downwind RTT configuration. The time axis is normalized with the blade edgewise natural frequency.

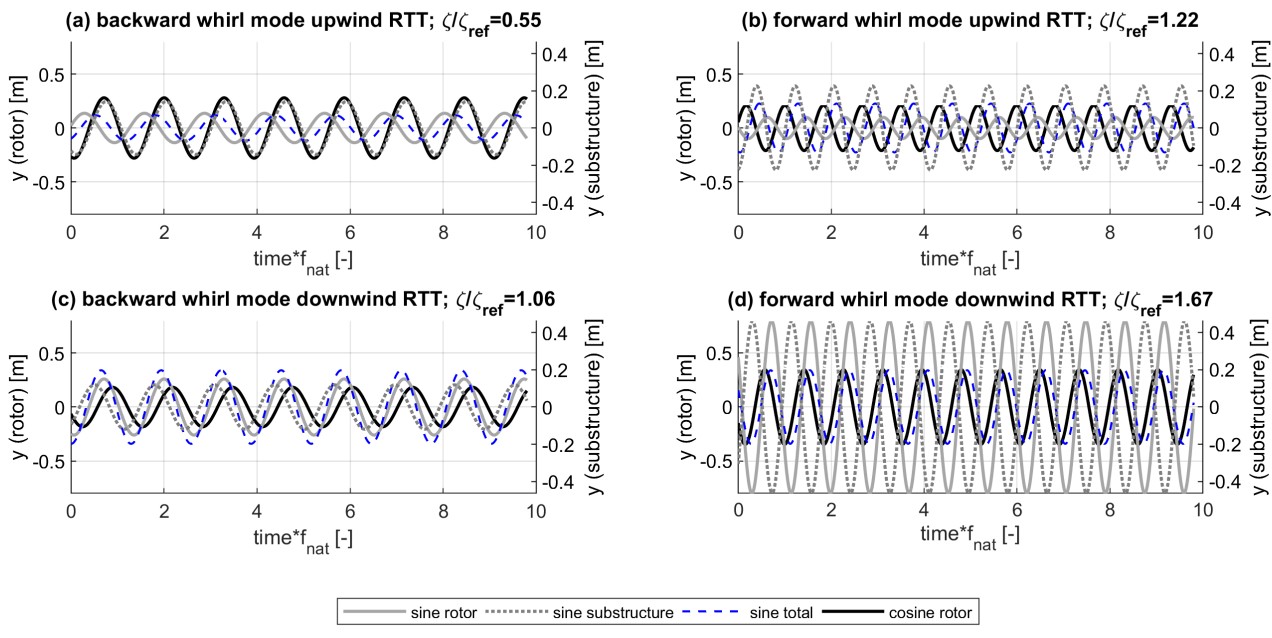

**Figure A3.** Modal out-of-plane displacements at $9\mathrm{ms}^{-1}$ and a tower torsional stiffness factor of 0.2 for the backward whirl mode and forward whirl mode of the upwind RTT and downwind RTT configuration. The time axis is normalized with the blade edgewise natural frequency.

*Author contributions.* GW set-up the models and carried out the calculations. LB and DV revised the models. All authors have interpreted the obtained data. GW prepared the paper with revisions of all co-authors.

*Competing interests.* This project is an industrial PhD project funded by the Innovation Fund Denmark and Suzlons Blade Science Center. Gesine Wanke is employed at Suzlons Blade Science Center.

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
