# Peer review of "Differences in damping of edgewise whirl modes operating an upwind turbine in a downwind configuration"

_Wind Energy Science, 2019_

## Referee Comment (RC1) · Anonymous Referee #1 · 22 Jan 2020

**General comments:** The paper deals with the study of the difference of in-plane

damping factors if an upwind turbine is operated in downwind configuration. Moreover, a trade study has been performed in order to evaluate the effect of some design parameters (i.e. shaft length, tower torsion and cone angle) on the damping factors.

Damping factors are estimated from suitable timeseries obtained from an aeroelastic code, using a simple data analysis tool (i.e. the logarithm decrement).

The paper is of interest for the wind energy community. Moreover, from the manuscript one may imagine that there should be a possible industrial appeal in trying to operate

an upwind rotor in downwind configuration. This is surely a plus of this work.

However, there some points to correct/clarify/improve so as to produce a manuscript which is worth publishing. These points are listed as "Major comments" here below.

**Major comments**:

- I strongly suggest enlarging the description of the procedure adopted to excite the whirl modes. In particular: is this already present in literature? Fore example there is something similar in "Thomsen et al, A Method for Determination of Damping for Edgewise Blade Vibrations. Wind Energy 2000, 3:233-246". Is there a proof (or reference) that the phase and the order of the harmonic excitation (120 deg with a sequence blade 1-2-3 and 3-2-1) are able to excite independently the backward and forward whirl modes? This is an important point, as the method chosen to the compute the damping (i.e. the log-decay) does not perform well if more than one mode is present in the measure. One ore more figure with the simulation outputs while they ring down (with possibly an FFT to check the effective harmonic content) to beinserted in the text (even if the methodology is well known) could be useful to demonstrate this fact.

- It is not clear the reason to show all the results of section 3.2, 3.3, 3.4 and 3.5 (very interesting results representing the core of the paper) for the RTT rotor (that is the rotor with reduced flexibility). In fact, it has been proved a big impact of tower torsion on whirl damping. Using a tower too rigid (or better unrealistically rigid) may lead to misleading results as the relative effects of other quantities (e.g. cone and shaft length) could appear more pronounced respect the most impacting quantity, that is the tower flexibility. Please comment.

- A turbine with few degrees of freedom has been used. As far as I have understood, the tower bending (fore-aft and side-side) flexibility are not present in the turbine model. This simplification seems a bit strong to study whirling charac-
teristics. I suggest clarifying this point, especially if Authors believe that such a simplification may alter the obtained results.

- Check in the entire text the correct spelling of "forward". It is often written as "foreward".

**Minor comments**

- Introduction, line 14-16. The sentence should be either rephrased or complemented by a reference. In fact, one may easily imagine that blade tip to tower clearance could be an active constraint also for downwind configurations in abnormal conditions involving shut-downs, where large forward blade deflections are to be expected. I understand that for downwind configurations this constraint may be somehow "relaxed", but I would consider "are not subject to such constraint" a bit too strong for such a complex problem.

- Line 50: "overall" instead of "over all".

- Line 76: The sentence "The turbine flexibility is reduced to the rotor flexibility and tower torsional flexibility . . ." is not clear. Please, rephrase.

- Line 85: Are coupling terms due to pre-bend and shaft tilt expected to play a prominent role in in-plane vibrations?

- Line 96: Setting the gravity to zero entails two significant effects. First, as written by the Authors, the periodic loading at rotor frequency caused by blade weight is nullified. Second, the periodic change in the blade stiffness is neglected. In fact, when a blade is upward, it compressed by its own weight, leading to a lower stiffness; on the other hand, the opposite happens when the blade is downward. This causes a periodic variation of the rotor/blade properties, which may have a significant effect on the turbine response.

- Line 104: "excited" instead of "exited"

- Section 2.2: The use of the Coleman transformation should be better explained:

  – Why only for 9 m/s?
  – The possibility to distinguish between the blade self-motion and substructure motion through the Coleman transformation is interesting but deserves additional descriptions. I may say that the two sources of vibration can be separated simply by looking at the spectra of the loads, as they should show up at two different frequencies. Hence, without the need of the Coleman transformation. But on this point, I may have missed something. . .
  – Line 123: I would use "Coleman" capitalized.
  – Line 126: "latter" instead of "later".
  – Line 129-133: The procedure adopted to guarantee the phase consistency needs additional explanation. Is it a standard practice?

- 'Results' section, figure 1 and line 140: Consider the possibility to write the normalization definition ("normalized with the damping of the upwind RTT configuration at 9 ms$-1$ of the forward whirl mode") also in the text to ease the reading.

- Line 154-156: "This indicates that the difference in damping is driven by the interaction of the aerodynamic forces on the rotor with the tower torsional motion". I consider this result important. Can the Authors say something about the tower bending motion? I expect whirl modes to be dependent also on the entire flexibility of the support structure, not only on the tower torsion.

- Line 155: "when the aerodynamic forces are not present, the damping of both forward and backward modes are identical": it is difficult to see it from the figure, as from the text I would expect two coincident lines.

- Line 159 (186): "cannot" instead of "can not".

- Line 174: "Overall"?

- Line 176: check the sentence, "displacements of to the substructure"

- Figure 2: The figure and the treatment seem interesting and important in the economy of the paper, but it should be better explained. How were those graphs generated? Are they results of the simulations, possibly transformed through the Coleman transformation? If so, why aren't they damped?

- Line 203: "From" instead of "form"

- Line 203: "is due to a difference in out-of-plane motion." It should be clarified whether it is a generic out-of-plane motion or the one entailed by tower torsion.

- Line 206: "The edgewise damping can be increased", to be checked. Do the Authors refer to the whirl damping or to the blade edgewise one? If I have understood correctly, it should be the latter.

- Section 3.2: in general, a good section. But what about the cosine components of rotor and substructure? Those components may generate out-of-plane vibrations contributing to the total damping of the whirl modes as well.

- Figure 3, 4 and 5: I would use "normalize whirl damping" avoiding the use of "edgewise damping" which may be misinterpreted with the blade edgewise damping.

- Chapter 4: In the model, only the tower torsion is considered. Do you expect that the tower bending may play a significant role? if not, why?

- Line 294-295. "From an edgewise damping point of view downwind configurations could benefit from towers with lower torsional stiffness". Very interesting results. Could a lower torsional stiffness have negative effects from some other points of view?

- Conclusion, line 308-309: The Authors mentioned only fore-aft tower bending. Is there a reason not to consider the tower side-side? For example, in [Bottasso and Cacciola, Model-independent periodic stability analysis of wind turbines, Wind Energy 2015] and [Allen, Sracic, Chauhan and Hanse, Output-only modal analysis of linear time periodic systems with application to wind turbine simulation data, Mech. Syst. Signal Pr. 2011] the whirl modes are clearly visible from tower side-side (also fore-aft) response.

---

## Referee Comment (RC2) · Vasilis A. Riziotis (Referee) · 26 Jan 2020

The paper compares the damping of an upwind 2.1 MW wind turbine against its downwind counterpart. Several variants of the upwind and downwind configurations are compared in the analysis. Emphasis is put on the assessment of the damping of the edgewise whirling modes (BW and FW) which are the lowest damped modes of the turbine. Aeroelastic frequencies and damping as well as mode shapes are directly calculated from simulated nonlinear analysis time-series. The main innovation of the present work is that the authors directly correlate the damping of the edgewise modes with the out-of-plane motion of the blades which is distinguished into their self

motion component but also the component due to the tower torsion. The paper is well structured and written and the results presented are interesting. Therefore, in the reviewer's opinion the paper deserves publication after some revision is made to the original text based on the comments below. 1) Not much is said about the method used in the assessment of the aeroelastic frequencies and damping. For example if it is based on peak values characterization on nonlinear time series (it seems to be so), the pros and cons of this simple approach should be highlighted in section 2.1 or 2.2. For example how good the predictions would be in case of multiple frequencies, closely spaced. The same holds for the mode shapes identification method. The authors are recommended to cite some references and to add some more discussion on the theoretical background of the methods used in their analysis. 2) It would be also nice to include some comparison of the results of the present nonlinear analysis method against the results of the linear eigenvalue analysis in a case in which HAWCstab2 simulations are valid (eg. standard upwind configuration). In particular the comparison of the predicted mode shapes is critical because much of the discussion that follows is based on these predictions. 3) Another point that it should be further elaborated in the conclusion section is to what extend the results obtained and the conclusions drawn can be generalized to any WT configuration or they are turbine specific. Additional minor comments that should be discussed by the authors as well as grammar/syntax corrections can be found in the accompanying pdf.

Please also note the supplement to this comment:
https://www.wind-energ-sci-discuss.net/wes-2019-88/wes-2019-88-RC2-supplement.pdf

**Supplement:**

[revised manuscript text omitted]

---

## Author Comment (AC1) · 17 Feb 2020

The answers to the referees as well as a manuscript with tracked changes are attached in the .pdf-file.

Please also note the supplement to this comment:
https://www.wind-energ-sci-discuss.net/wes-2019-88/wes-2019-88-AC1-supplement.pdf

---

## Author Response (AR1)

**Answer to referee comments of referee 1**

Dear Referee #RC1,

Thank you very much for your detailed feedback. It is very much appreciated and enhances our work!

*Major comments*

**Excitation method:** It is not very common to excite the blades in the blade coordinate system, simply because it is hard or even impossible to achieve in an experimental set-up. We have chosen this method, because we could achieve very clear spectra which is crucial for the damping estimation from peak-to-peak counting. Example spectra and a paragraph are. An example of the decaying time-series is also added together with the resulting estimated damping and the restrictions of the used method.

**Analysis based on turbine with reduced flexibility:** The reason to base the analysis on the RTT rotor is that with it is possible to capture the difference in edgewise whirl damping between the two rotor configurations (upwind and downwind) with the minimum degree of freedom. The fully flexible turbine has a lower damping especially for the forward whirl mode. However, these additional degrees of freedom have a similar impact on both, the upwind and the downwind configuration. Thus, these are not a causation of difference in edgewise loads that can be observed in load simulations.
The advantage of the reduced degrees of freedom is, that it allows to separate the motion of the substructure from the rotor motion without influence of interaction of the nacelle or shaft. This might on the other hand over pronounce the impact of the shaft length or the cone angle. However, it does show the trends and the difference in interaction of the rotor with the tower torsion which has been proven to have a big impact on the edgewise whirl damping. It should also be pointed out, that the absolute edgewise whirl damping is very turbine specific. Even with the full flexibility, this work shows the trends of change when changing the turbine configuration from upwind to downwind, and how the difference in damping can be influenced. Thus, the plots of the parameter variations cannot be compared to each other regarding absolute damping values.
A comment to this is added in the text for clarification.

**Study of whirl characteristics with reduced degrees of freedom:** As we have shown from the comparison of the fully flexible turbine configuration with the turbine configurations with reduced number of degrees of freedom, the difference in damping between the upwind and the downwind configuration can be captured. Including all degrees of freedom does change the whirl characteristics, especially through the tower flexibility. However, we could only observe small differences of the influence on the upwind configuration compared to the downwind configuration which is what this work focusses on.
Generally, this study should be extended in future work to get the full picture of the difference of whirl characteristics.
A comment to this is added in the discussion section.

**Forward spelling:** has been corrected.

*Minor comments*

**Line 14-16:** It is true, that also downwind turbines can be constraint in blade design depending on the geometrical configuration of the turbine as well as the design driving situations and the handling of fault situations by the controller. The introduction is changed accordingly.

**Line 76-78:** (The turbine flexibility is reduced to the rotor flexibility and tower torsional flexibility, as this configuration resembles the difference in edgewise damping with the minimum degrees of freedom.) The sentence is rephrased to: The resulting configuration consists of a fully flexible rotor and a tower that is only flexible in torsion. This configuration with reduced flexibility captures the main difference in edgewise whirl mode damping and allows a more detailed analysis of modal displacements.

**Line 85:** (coupling terms due to prebend and shaft tilt) The coupling terms are not expected to play a prominent role in the in-plane vibrations (This has been checked with hawcStab2 for the upwind configuration). They are neglected rather to avoid excitation of the out-of-plane modes from the forces that are used for excitation. Otherwise, the peak-to-peak counting used for damping estimation and especially the mode shape analysis are misleading. A comment on this has been added to the manuscript.

**Line 96:** (By setting gravity to zero, the variation of stiffness is neglected) It is assumed that the structural stiffness and the centrifugal effect are dominant and the variation of stiffness from gravity is negligible. However, this cannot be quantified by the chosen method as the rotation in the gravity field would permanently excite the in-plane blade motion which makes the damping prediction from the time-series impossible. HAWCStab2 as a tool is no alternative since it neglects the effect of gravity. Further analysis of this effect could be done with Floquet-analysis, but this is outside the scope of this work.

**Section 2.2:** (why only 9m/s?): 9m/s is chosen as the observed difference in damping so differences in modal displacements are expected to be largest.
(Distinguishing between self-motion and substructure motion): To be able to investigate the modal displacement the displacement of the blades is extracted in the blade coordinate system, while the displacement due to tower motion is extracted in the tower coordinate system. To be able to do the analysis of the contribution to out-of-plane motion all displacements need to be in the same coordinate system, which requires the Coleman-transformation. For the damping estimation, this would not be required. The text is adjusted for clarification.
(Procedure for phase consistency) This procedure is not standard practice. However, phase consistency has to be assured, as several spectra are calculated for each signal independently (blades and tower torsion). The implemented Matlab procedure gives a spectrum with relative phases for each spectrum. Further explanation is added.

**Line 140:** normalization definition is added.

**Line 154-156:** "Other degrees of freedom such as tower bending flexibility do influence the absolute damping of the whirl modes but do not cause the major differences between the upwind and the downwind configuration." Added as a comment in the text.

**Line 155:** The coinciding lines are not plotted and this is pointed out in the text as additional information.

**Figure 2:** The time-series are regenerated from the spectrum of the time-series to assure the displacement is a modal displacement. The damping is therefore not included. The information is added in the description of the method.

**Line 203**: Generic out-of-plane motion or due to the tower torsion? The "total" out-of plane motion is added to the text.

**Line 206**: the edgewise "whirl" damping is added to the text.

**Section 3.2:** The text is referring to the cosine component of the rotor contributing to 5% of the difference in the out-of-plane displacements (line 182-183) The substructure does not have a contribution to the cosine displacement since the tower bending is neglected.

**Figure 3-5:** "edgewise damping" has been replaced by "edgewise whirl damping"

**Chapter 4 & Line 308-309: "**Expected role of tower bending" expanded in section 5. The tower fore-aft bending is expected to be seen directly in the out-of-plane displacement. The tower side-side bending couples directly to the asymmetric edge whirl motion and can further be expected to couple to out-of-plane displacement through the yaw motion, as the center of gravity of the rotor nacelle assmbly is not at the tower center.

**Line 204-205:** total impact of low tower torsional stiffness on downwind configurations has not yet been investigated. For upwind configurations the lower edgewise damping for lattice structures is well known. Further investigations would need to be made and full load assessment would be required to fully evaluate the effect of the lower tower torsional stiffness.

**Generally:** Spelling and grammatical comments are implemented.

**Answer to referee comments of referee 2**

Dear Vasilis A. Riziotis,

Thank you very much for your detailed feedback. It is very much appreciated and improves our work!

***Main comments***

1.) Pros and Cons for damping assessment are added in the text. It is shown that there is only one frequency dominant in the time-series and examples are given. The generation of mode shapes is described in more details.
2.) A comparison of HAWC2 and HAWCStab2 for frequency and damping are added. Mode shapes could not be validated. The required mode shapes were not an official output from HAWCStab2 and needed post-processing from a binary file. It was possible to extract the amplitudes, but the phases could not be converted correctly for both edgewise whirl mode, even for the upwind configuration. The mode

shapes from HAWC2 on the other hand could be extracted in known coordinate systems and seemed to be consistent. However, there is no doubt that more work is required regarding a validation of the mode shapes.

3.) We checked that the DTU-10MW-RWT shows the same trend for the damping of the two edgewise whirl modes. However, how strongly this is pronounced is turbine specific. A comment is added in the discussion to point out to what extend the conclusions are turbine specific.

**Supplement comments**

**Line 34:** (Could you please become more specific. You are probably referring to bringing close to each other FW edgewise mode with the BW 2nd flap mode.) The 1$^{st}$ FW edge and 2$^{nd}$ BW flap mode have been added in the text.

**Line 76-78:** (The turbine flexibility is reduced to the rotor flexibility and tower torsional flexibility, as this configuration resembles the difference in edgewise damping with the minimum degrees of freedom.) The sentence is rephrased to: The resulting configuration consists of a fully flexible rotor and a tower that is only flexible in torsion. This configuration with reduced flexibility captures the main difference in edgewise whirl mode damping and allows more detailed analysis of modal displacements.

**Line 97-98:** (The controller is exchanged by a simple setting of pitch angle and rotational speed according to the wind speed at hub height to allow for a slow wind speed increase to avoid other modal frequencies than the excited frequencies in the timeseries.) Sentence is rephrased to: "[…] to avoid the excitation of modal frequencies other than the edgewise whirl frequencies. This resembles a fix-free drive train operational mode." The collective edgewise mode is not excited due to the chosen excitation method and the gravity set to zero. This has also been checked from the signal in the frequency domain. Otherwise, the estimation of damping from the time-series would be misleading.

**Line 106-108:** (It has been tested with the aeroelastic modal analysis tool HAWCStab2 (\cite{Hansen2004}) that the trends over wind speed as well as trends for the difference of damping between the configurations are captured correctly for investigations of qualitative differences. "This was done for the validation of the nonlinear damping characterization method and I suppose it was performed for the upwind turbine. Is that correct?") This has been done for validation of the nonlinear damping characterization method of both upwind and downwind configuration. HAWCStab2 agrees with the change of damping over wind speed, as well as, the ratio of damping between the upwind and the downwind configuration. However, the absolute values are not the same and more investigations are needed.
We have also tried to validate the mode shapes between HAWC2 and HAWCStab2, but the phases do not agree. We assume, that this is an issue of the output file generated by HAWCStab2, as the sorting of mode shapes and coordinate-systems of the auto-generated data could not be clarified, especially for the downwind configuration.

**Line 109-111:** (For a primary damping estimation the damping coefficient for a single airfoil as described by Hansen (2007) is calculated.) Text is changed for clarification of the aim, namely to assure that differences of damping are not due to operational points of the airfoil and/or the steady states of the turbine configurations.

**Line 127-132:** (Calculation of modal displacement and phase consistency). The section has been extended as it caused confusion. No direct system identification method has been used. It has been assured, that the method of excitation only excites one of the whirl modes, which has been checked from the spectra. Mode shapes are then calculated from the frequency and phase information of the spectra. An additional explanation has been added. The method of phase consistency has been described in more detail as there is no reference and it might not be a common standard.

**Line 134:** Validation for the frequency and damping has been added. The mode shapes could not be validated, due to unresolved issues with the output from HAWCStab2. (The output does not agree with the internal animation inside HAWCStab2.) The latter is the main reason to finally do the study with HAWC2.

**Line 140:** normalization definition is added.

**Line 289:** A paragraph is added to explain to what extend the conclusions and results are turbine specific.

**Generally:** Spelling and grammatical comments are implemented.

[revised manuscript text omitted]

Configurations and parameter variations configuration/ parameter variation propertiesedgewise damping estimation all configurations no tilt, no cone, no prebendsimplified controller, no gravity, uniform inflow(no turbulence, no sheer, no veer, no inclination angle)upwind FF upwind, all degrees of freedom (fully flexible)downwind FF downwind, all degrees of freedom (fully flexible) upwind RTT upwind, rotor flexibility, tower torsion flexibilitydownwind RTT downwind, rotor flexibility, tower torsion flexibilityparameter variation shaft length up- and downwind RTT configurationshaft length variation: -30% to +100% cone angle up- and downwind RTT configurationcone angle variation: 0° to 7.5° (away from tower)tower torsional stiffness up- and downwind RTT configurationtorsional stiffness factor variation ± 80%

---

## Referee Report (RR1)

Second review of the paper entitled "Differences in damping of edgewise whirl modes operating an upwind turbine in a downwind configuration"

The manuscript, in this new version, improved as I expected. In particular, the new introduced parts and figures related to the excitation method are exactly what I expected to see to demonstrate the goodness of the proposed approach. Moreover, the demonstration that the employed perturbation can separately excite the whirling modes represent a good value for the paper (at least in my opinion).

Minor technical comments:

- By setting gravity to zero, the variation of stiffness is neglected: The Authors wrote in the reply "Further analysis of this effect could be done with Floquet-analysis, but this is outside the scope of this work.". I fully agree with this. Please, consider the possibility to add somewhere in the text this point.

- The procedure for the phase consistency check is still a bit confusing. Probably, I haven't understood the need for this procedure. In fact, if one records different signals in the very same instants of time (with the same time step and the same known initial time), the phases computed by the Fourier Transformation results to be consistent. In a simulation environment, this can be easily done. In a real environment, probably, the phase consistency is an issue to solve when different sensors with different sampling time are employed. Is this procedure borrowed from real field applications, even though in silico it is not strictly necessary?

- Check sentence: "Further investigations should evaluate the influence of the upper especially on the mode shapes of the whirl modes".

---

## Author Response (AR2)

Dear editor and reviewer,

Thank you for the comments. We have checked and implemented the following

*- By setting gravity to zero, the variation of stiffness is neglected: The Authors wrote in the reply "Further analysis of this effect could be done with Floquet-analysis, but this is outside the scope of this work.". I fully agree with this. Please, consider the possibility to add somewhere in the text this point.*
Implemented at line 104-105

[revised manuscript text omitted]